# Robust Symbolic Regression for Dynamical System Identification

**Ramzi Dakhmouche** *ramzi.dakhmouche@epfl.ch*
*Institute of Mathematics, EPFL*
*Laboratory for Computational Engineering, EMPA*

**Ivan Lunati** *ivan.lunati@empa.ch*
*Laboratory for Computational Engineering, EMPA*

**Hossein Gorji** *Mohammadhossein.Gorji@empa.ch*
*Laboratory for Computational Engineering, EMPA*

**Reviewed on OpenReview:** *https://openreview.net/forum?id=ZfPbCFZQbx&nesting=2&sort=date-desc*

## Abstract

Real-world complex systems often miss high-fidelity physical descriptions and are typically subject to partial observability. Learning the dynamics of such systems is a challenging and ubiquitous problem, encountered in diverse critical applications which require interpretability and qualitative guarantees. Our paper addresses this problem in the case of sparsely observed probability distribution flows, governed by ODEs. Specifically, we devise a *white box* approach -dubbed Symbolic Distribution Flow Learner (`SDFL`)- leveraging symbolic search with a Wasserstein-based loss function, resulting in a robust model-recovery scheme which naturally lends itself to cope with partial observability. Additionally, we furnish the proposed framework with theoretical guarantees on the number of required *snapshots* to achieve a certain level of fidelity in the model-discovery. We illustrate the performance of the proposed scheme on the prototypical problem of Kuramoto networks and a standard benchmark of single-cell RNA sequence trajectory data. The numerical experiments demonstrate the competitive performance of `SDFL` in comparison to the state-of-the-art.

## 1 Introduction

Complex systems often emerge as networks of physical and societal interactions, with examples ranging from epidemics to consensus dynamics, and from power grids to biological organisms. The pursuit of accurate symbolic expressions that describe the evolution of such dynamical systems, is thus of paramount importance in many areas of science and engineering. Indeed, such parsimonious model descriptions offer several advantages, including compactness, explicit interpretations and high-fidelity generalization. Unlike *black-box* approaches, they give explicit insights on the underlying processes. Most notably, they allow for safety related qualitative guarantees such as asymptotic stability, which are crucial for critical applications.

Yet complex systems are prone to partial observability, uncertainty and presence of noise, which render deriving explicit representations particularly challenging. Hence, devising robust and efficient learning methods to uncover equations directly from data -also known as symbolic regression- is of great interest.

Symbolic regression has been extensively studied over recent years in both algebraic and differential equation discovery contexts. Common approaches include: sparse regression (Brunton et al., 2016; Rudy et al., 2017; Chen et al., 2021; Kubalík et al., 2023), sequence-to-sequence deep neural network modeling (Becker et al., 2022; Vastl et al., 2022; Biggio et al., 2021), and symbolic search-based formulations (Cornforth

& Lipson, 2012; Gaucel et al., 2014; Cazenave, 2013; Lu et al., 2021; Sun et al., 2023). One important setting, which has not been studied much though, is the discovery of dynamical systems governing probability distribution flows using only a few sampled screen shots, which is particularly relevant for many applications ranging from epidemics modeling to cellular evolution prediction (Bunne et al., 2022). Consequently, we propose to address this question, in the case of network flows, by designing a suitable equation recovery framework and illustrating its numerical performance on both synthetic and real-world data. This setup of symbolic learning of probability flows in networks entails several challenges:

- *Snapshots*, arriving at random times, have to be described in a continuous-time probabilistic representation.

- A robust inference algorithm enabling efficient search for an accurate solution -under partial observability- has to be designed.

- Permutation invariance, which is inherent in network structures, should be incorporated into the symbolic search, in particular to reduce the computational cost.

These challenges are addressed by our technical contributions:

1) We propose a trajectory inference (Bunne et al., 2022) approach based on symbolic search, yielding *white-box* models with the permutation invariance property,

2) We devise a suitable loss function that leverages the robustness properties of the Wasserstein distance (Villani, 2009), while taking into account the limited observability of the system under study,

3) We back our devised algorithm with suitable sample complexity theoretical results,

4) We demonstrate the performance of the proposed approach on the prototypical problem of Kuramoto networks and a standard benchmark of single-cell population trajectory data. Our approach handles settings with low sampling frequency of the order of 100 times less for the Kuramoto model (see Appendix E.4) than what is required by the current symbolic regression state-of-the-art (Qian et al., 2022; Sun et al., 2023; Gao & Yan, 2022).

The rest of the paper is organized as follows. In section 2, we review related works and partially motivate our design choices. In section 3, we introduce the notions upon which the proposed algorithm is based. In section 4, we outline the main contributions of this work and provide the theoretical results that guided the design. In sections 5, we present numerical evaluations, followed by concluding remarks in section 7.

## 2 Related works

**Symbolic search.** Symbolic regression was initially formulated as a discrete optimization problem (Augusto & Barbosa, 2000; Smits & Kotanchek, 2005; Cornforth & Lipson, 2012; Cazenave, 2013), where the goal is to find the most accurate mathematical expression based on a predefined set of elementary operations and functions (e.g. $+, -, \times, \sin, \exp, \dots$). The mathematical expressions were represented through a one-to-one correspondence with *pre-order traversal* trees. For the resolution, Genetic Programming (GP) heuristics (Koza, 1994) were used to recover the equations underlying the training data. This method subsequently inspired applications in population evolution modeling (Bongard & Lipson, 2007), prediction of solar power production (Quade et al., 2016), and Eulerian fluid flow hidden parameterization discovery (Vaddireddy et al., 2020). Yet, GP suffers from a number of issues including over-fitting, brittleness to noise and poor-scalability (Brunton et al., 2016; Petersen et al., 2019; Lu et al., 2021). A more recent approach (Petersen et al., 2019) uses a Deep Reinforcement Learning (DRL) model to solve the optimization problem and generally outperforms GP-based models. However, deep learning models require large amounts of data in addition to their relative lack of full-automation, since architecture hyper-parameters have to be

tuned by a human expert. On the other hand, a parallel well performing approach with the potential to achieve full-automation is Monte-Carlo Tree Search (MCTS). Based on the tree representation used in GP formulations, MCTS builds upon exploration-exploitation trade-off insights from sequential learning (Munos et al., 2014). It is designed for a stochastic setting where data-points are costly to obtain and therefore naturally handles noisy input and scare-data settings well. Some works have applied MCTS successfully to simple (Cazenave, 2013; Islam et al., 2018; Lu et al., 2021) and broader (Sun et al., 2023) symbolic regression problems. The former used MCTS to uncover non-linear expressions in a supervised setting, whereas the latter also applied it to Ordinary Differential Equation (ODE) discovery when the initial conditions are deterministic. In the case of network structured problems, (Shi et al., 2023; Cranmer et al., 2020) are the only works -to the best of our knowledge- which distill explicit equations. That is done by training a Graph Neural Network (GNN) and distilling equations for the message-passing operators. Additionally, only deterministic physical systems for which the whole trajectory is observed are considered. In contrast, we address a setting, where the goal is to identify the governing equations of *stochastic* network dynamics (with randomness in the initial condition) from the observation of a reduced number of screen shots across time.

**Sparse regularization.** One common property that emerges when modelling many natural phenomena and engineering problems is sparsity. Leveraging this fact, a number of works (Brunton et al., 2016; Schaeffer, 2017; Rudy et al., 2017; Loiseau & Brunton, 2018) formulate (deterministic) dynamics discovery as a $\ell_1$-regularized linear regression problem, over a predefined dictionary of basis functions. The resulting convex optimization is then solved using a sequential thresholding scheme. In particular, Brunton et al. (2016) demonstrate the performance of the scheme they propose, dubbed SINDy, uncovering the chaotic Lorenz system and a fluid vortex shedding ODE. Subsequently, Boninsegna et al. (2018); Huang et al. (2022) extend the approach to the discovery of Stochastic Differential Equations (SDEs), using the Kramers-Moyal formula. Besides that, Chen et al. (2021) addresses a scarce-data setting by training a neural net as a surrogate, combining data from PDEs with various boundary conditions. Once the neural network is pretrained, it is used to discover the underlying PDE using an alternating direction optimization (ADO) algorithm to solve the $\ell_1$-regularized linear regression problem. From a theoretical perspective, Schaeffer et al. (2018) study the question of minimal number of screen-shots required to recover multivariate quadratic ODEs, in the case of random initial condition. However, they assume velocities to be known which is inconsistent with the small and scarce data setting. Overall, although usefulness of sparsity-promoting approaches has been extensively demonstrated, they still rely on prior knowledge to define their main component, namely the basis function library. Whereas, if a library of massive size is chosen, the algorithm empirically fails to hold the sparsity constraint (Sun et al., 2023). In comparison, we apply MCTS for our symbolic search part, which is not bound by such constraints.

**Sequence to sequence models.** The most recent approach to symbolic regression leverages the success of deep learning in sequential data modeling, such as in natural language processing (Devlin et al., 2018). More precisely, (Biggio et al., 2021; Kamienny et al., 2022; Vastl et al., 2022) propose a transformer-based architecture, which is trained to output mathematical expressions based on data-sets of feature-prediction pairs. That is, for each input data-set $\{(x_i, y_i)\}_{i=1}^n$, the model outputs an expression $e$ corresponding to a function $f_e$ satisfying $\forall i \in [1, n]$, $y_i \simeq f_e(x_i)$. One challenge though, is to generate a training data-set $\left\{ (\{(x_i^j, f_{e_j}(x_i^j))\}_{i=1}^n, f_{e_j}) \right\}_{j=1}^N$ which is rich enough to represent parsimonious equations that are frequently encountered in practice. To achieve that, equations are generated as binary trees, following the work of (Lample & Charton, 2020; Kusner et al., 2017) and previously mentioned symbolic search approach representations, e.g. (Cazenave, 2013). On the other hand, (Li et al., 2019) combine a recurrent neural network with MCTS to enforce asymptotic constraints on the learned expression, while some works (Martius & Lampert, 2016; Sahoo et al., 2018; Costa et al., 2020; Kubalík et al., 2023) propose to use different elementary mathematical operators (e.g. $+, \times, \cos \dots$ etc) as activation functions for neural net architectures, while imposing sparsity on the parameters to extract interpretable analytic expressions. However, these approaches rely on the generation of large data-sets, which is prohibitively costly for high-dimensional ODE/PDEs.

**Neural ODEs.** First introduced (Chen et al., 2018) as a continuous counterpart to residual net-

works for generative modeling, neural ODEs have demonstrated success in several tasks such as density estimation, image generation and variational inference (Grathwohl et al., 2018). They are based on modeling the vector field defining an ODE as a neural network and leveraging the adjoint sensitivity method (Pontryagin, 2018) for efficient back-propagation. Subsequently, they have been extended to stochastic and latent variable settings allowing for an increased expressive power (Dupont et al., 2019; Kidger, 2022). Besides, notable parallel enhancements included extensions to non-Euclidean spaces (Lou et al., 2020; Jeong et al., 2023) by devising a manifold adjoint method. However, one of the main drawbacks of neural ODEs is the fact that they do not allow for qualitative analyses of the models, which is often needed for physical and biological systems.

## 3 Problem Formulation and Background

### 3.1 General setup

Fix $T > 0$ as a positive time horizon, and consider the random variable $x_0 \in \mathbb{R}^d$ to be the initial condition of the state variable $x_t$ which evolves according to the ODE system

$$\dot{y} = f(y) \tag{1}$$

in time $t \in [0, T]$. In our setting, $f$ encodes the interactions between different components of $x_t$ over time, which are distributed according to a known and fixed network topology $\mathcal{G}$. We denote by $\mu_t$, the probability measure induced by $x_t$ i.e. $x_t \sim \mu_t$, and by $p_t$ the corresponding probability density. The latter evolves -if $\mu_t$ is absolutely continuous with respect to the Lebesgue measure- according to the equation

$$\log p_t(x_t) = \log p_0(x_0) - \int_0^t \mathrm{Tr}\left(f'(x_s)\right) ds$$

along the trajectory $(x_s)_{s \in [0,T]}$ (see e.g. Chen et al. (2018)). For $n \geq 2$ measurement times $\{t_0, t_1, \ldots, t_n\}$, assume $m \geq 1$ samples of each of the distributions $\mu_{t_0}, \mu_{t_1}, \ldots, \mu_{t_n}$ are observed, resulting in *snapshots* represented by the corresponding empirical probability measures $\{\hat{\mu}_{t_1,m}, \ldots, \hat{\mu}_{t_n,m}\}$. The question addressed in this paper is that of the recovery of the function $f$ defining the ODE that governs the given dynamics. That is, *trajectory inference* (Hashimoto et al., 2016; Tong et al., 2020; Bunne et al., 2022; Huguet et al., 2022), through an explicit closed form ODE. For that matter, the solution of the ODE can be seen as the image of $f$ by the operator given by

$$F(f) : (t, x_0) \mapsto F^t(f)(x_0)$$

where $t \mapsto F^t(f)(x_0)$ is the solution of the Cauchy problem:

$$\begin{cases} \dot{y} = f(y) \\ y(0) = x_0 \end{cases} \tag{2}$$

for a given initial condition $x_0 \in \mathbb{R}^d$. Note that, the problem of recovering $f$ from $(\hat{\mu}_{t_i,m})_{i \in \{1,\ldots,n\}}$ is *ill-posed* in general, and further assumptions are needed. Such assumptions should represent prior information that can act as a form of regularization rendering the learning task at hand more feasible. In our case, the required prior information comes from the fact that the estimator of $f$ is constructed by combining analytic expressions from a fixed pre-selected set. We make the assumption that the observation instants fulfill $\forall i \in \{0, \ldots, n\}, t_i \in (iT/n, (i+1)T/n)$. A straight-forward extension, though, can be obtained for uniformly sampled time instants $(t_i)_{i \in \{0,\ldots,n\}}$ based on a Quasi-Monte Carlo scheme convergence argument (Niederreiter, 1978). To translate this setting to an optimization framework, first, we review some concepts about distances in probability spaces upon which a suitable goodness-of-fit measure is proposed.

### 3.2 Wasserstein guidance

Considering probability distribution flow modeling, we will need to compare predicted distributions with observed ones. Popular measures of disparity between probability distributions include the Kullback–Leibler

(KL) divergence in the computational context, the total variation distance in the theoretical one, and the Wasserstein distance (Villani, 2009) in both. Indeed, due to its singular theoretical properties, e.g. weak topology metrization, convexity and robustness (see e.g. Mohajerin Esfahani & Kuhn (2018) for the latter), but also its amenability to efficient computation, the 2-Wasserstein distance $\mathcal{W}_2$ represents a natural choice for the design of a robust trajectory inference algorithm. Consider two measure spaces $(\mathcal{X}, \mu)$ and $(\mathcal{Y}, \nu)$ and denote by $\Pi(\mu, \nu)$ the set of their couplings, then $\mathcal{W}_2$ reads

$$\mathcal{W}_2(\mu, \nu) = \min_{\pi \in \Pi(\mu,\nu)} \int_{\mathcal{X} \times \mathcal{Y}} \|\alpha - \beta\|_2^2 \, \mathrm{d}\pi(\alpha, \beta)$$

where $\| \, . \, \|_2$ is the Euclidean norm. If we had access to $(\mu_t)_{t \in [0,T]}$, the goodness-of-fit of a candidate estimator $\hat{f}$ could be suitably defined as:

$$L(\hat{f}) = \int_0^T \mathcal{W}_2(F^t(\hat{f})_\# \mu_0, \, \mu_t) \, \mathrm{d}t \tag{3}$$

where $F^t(\hat{f})_\# \mu_0$ is the push-forward of the initial probability measure by the partial flow map $x_0 \mapsto F^t(\hat{f})(x_0)$. In other words, $L(\hat{f})$ is the time aggregated Wasserstein distance between the measure resulting from the inferred trajectory and the one resulting from the ground-truth dynamics. However, since $\mu_t$ is only partially known, we work with an approximation of $L(\hat{f})$ based on the observed *snapshots*, as discussed in the following, after presenting the basic building blocks of MCTS. Note that, the KL-divergence could not be used here given that it assumes that the compared distributions have the same support.

### 3.3 Monte-Carlo Tree Search for Symbolic Search

Unlike genetic programming, which suffers from over-fitting and brittleness to noise, MCTS handles stochastic settings by design in a data-efficient way (see for e.g. (Sun et al., 2023; Lattimore & Szepesvári, 2020)). The latter property also sets it forth compared to reinforcement learning approaches, which require large amounts of data. Due to the combinatorial nature of the state of possible mathematical expressions, it is crucial to reduce the number of required evaluations. To cope with that, MCTS targeted to symbolic search, relies on tree representations of mathematical expressions, where the nodes express unitary $(\sin(\cdot), \cos(\cdot), \dots)$ or binary operations $(+, -, \times, \dots)$ and leafs express variables and constants $(x_1, x_2, c, \dots)$. This is formalized using the notion of context-free grammars (Kusner et al., 2017; Sun et al., 2023). Given such representation, the intuitive idea behind MCTS is to explore as few operators as possible in building the tree, yet to identify the optimum with a high probability. To achieve that, it leverages insights from Upper-Confidence Bound (UCB) algorithms in sequential learning (Lattimore & Szepesvári, 2020), where the goal translates to optimizing a function which measures both accuracy and under-exploration of a candidate operation. We refer to (Sun et al., 2023) for a detailed pseudo-code, and recall the main components translated to our context below:

**Stochastic roll-outs.** To complete a partially built tree i.e. a partial expression denoted by $s$, the value of each potential additional node $a$ (representing an elementary operation) needs to be obtained, in order to choose the one with the highest value. For that matter, for each choice of node, random roll-outs of potential completions are performed. Each roll-out will either lead to a complete expression $\hat{f}(x)$, whose corresponding score function can be computed; or to an expression whose length exceeds a predefined maximal length $M$ leading to a score of 0. The largest of the obtained scores is then stored in $V(s, a)$ to be used in the next step.

**Exploration-aware selection.** Once all operations have been visited at least once, select the next one by maximizing a performance measure that takes into account under-exploration, and which can be taken in practice [1] as

$$UCT(s, a) := V(s, a) + c\sqrt{\ln[N(s)]/N(s, a)}, \ c > 0$$

---

[1] For the sample complexity result presented in prop. 1 to hold though, the scheme should be based on the more involved expression for UCT described in (Shah et al., 2020).

where $V(s, a)$ is the maximal value of operation $a$, given the operations constituting the current tree i.e. $s = [a_0, a_1, \ldots, a_{p-1}]$. Furthermore, $N(s, a)$ is the number of times that operation $a$ was chosen at tree state $s$ with $N(s) = \sum_{a \in \mathcal{A}(s)} N(s, a)$, where $\mathcal{A}(s)$ is the set of available operations at tree state $s$. Note that the square root term $\sqrt{\ln[N(s)]/N(s, a)}$ quantifies the under-exploration of operation $a$ at state $s$.

# 4 Technical Approach

To minimize the introduced loss function, given by equation 3, we replace the neural net estimator in the neural ODE framework of (Chen et al., 2018; Grathwohl et al., 2018; Kidger, 2022) with a search for an explicit analytic formula for $f$. This is achieved through a symbolic regression algorithm instantiated by MCTS (Sun et al., 2023). The devised workflow -of inferring the network dynamics from given snapshots- is illustrated in Fig. 6. It is based on selecting a given ODE candidate, evaluating its performance, and using that evaluation to inform future selections, as presented below. From a guarantee perspective, in addition to a bound on the approximation error of $L(\hat{f})$ by its empirical estimate $\hat{L}_{m,n}(\hat{f})$ for a given $\hat{f}$, we quatify the sample complexity of MCTS, using regret bound-based results from bandit theory (Lattimore & Szepesvári, 2020).

## 4.1 Discrete Loss function

The convergence of the discrete version of the loss function, defined for continuously differentiable $\hat{f} : \mathbb{R}^d \to \mathbb{R}^d$ by

$$\hat{L}_{m,n}(\hat{f}) = \frac{1}{n} \sum_{i=1}^{n} \mathcal{W}_2(F^{t_i}(\hat{f})_\# \hat{\mu}_{t_0,m}, \hat{\mu}_{t_i,m}), \tag{4}$$

to the continuous one i.e. $L$, is controlled by the number of snapshots and the size of the snapshot sample set. We build upon results of Fournier & Guillin (2015); Bonnans & Shapiro (2013) about *finite-sample* rates of convergence of the empirical measure in the Wasserstein space, and regularity of the Wasserstein distance to obtain the following theorem, where the proof is postponed to Appendix A.

**Theorem 1.**
Let $(\mu_t)_{t \geq 0}$ have a compact support and $\hat{f} : \mathbb{R}^d \longrightarrow \mathbb{R}^d$ be a differentiable function. Assume that $t \mapsto \mu_t$ is differentiable. Then, for all $m, n \geq 2$,

$$\mathbb{E} \left| \frac{1}{n} \hat{L}_{m,n}(\hat{f}) - \frac{1}{T} L(\hat{f}) \right| = O \left( \frac{1}{m^{(d/2+1)}} + \frac{1}{n} \right) ,$$

where $\hat{\mu}_{t_i,m}$, for $i \in \{1, \ldots, n\}$, denotes the empirical measure corresponding to the $m$ observed realizations of each of the probability distributions $(\mu_{t_i})_{1 \leq i \leq n}$.

*Proof.* We sketch the proof here and postpone the more detailed version to Appendix A. Without loss of generality, we assume $T = 1$. By theorem 1 in (Fournier & Guillin, 2015), given that $x_0 \mapsto F^t(x_0)$ is continuous and for all $i \in \{1, \ldots, n\}$, $\mu_{t_i}$ has compact support on the Polish space $\mathbb{R}^d$, there exists $C_1, C_2 > 0$ such that

$$\mathbb{E} \left[ \frac{1}{n} \sum_{i=1}^{n} \mathcal{W}_2(F^{t_i}(\hat{f})_\# \hat{\mu}_{t_0,m}, F^{t_i}(\hat{f})_\# \mu_{t_0}) \right] \leq \frac{C_1}{m^{(d/2+1)}}$$

$$\text{and} \qquad \mathbb{E} \left[ \frac{1}{n} \sum_{i=1}^{n} \mathcal{W}_2(\hat{\mu}_{t_i,m}, \mu_{t_i}) \right] \leq \frac{C_2}{m^{(d/2+1)}} .$$

Therefore, thanks to the triangular inequality satisfied by $\mathcal{W}_2$, we get

$$\mathbb{E} \left| \frac{1}{n} \hat{L}_{m,n}(\hat{f}) - \frac{1}{n} \sum_{i=1}^{n} \mathcal{W}_2(F^{t_i}(\hat{f})_\# \mu_{t_0}, \mu_{t_i}) \right| \leq \frac{C_1 + C_2}{m^{(d/2+1)}} .$$

On the other hand, by the stability property of optimal transport as shown in Appendix A, we get

$$\left| L(\hat{f}) - \frac{1}{n} \sum_{i=1}^{n} \mathcal{W}_2(F^{t_i}(\hat{f})_{\#}\mu_{t_0}, \mu_{t_i}) \right| \leq \frac{C_3}{n} \ ,$$

leading to

$$\mathbb{E} \left| \frac{1}{n} \hat{L}_{m,n}(\hat{f}) - L(\hat{f}) \right| = O\left( \frac{1}{m^{(d/2+1)}} + \frac{1}{n} \right).$$

$\square$

## 4.2 Symbolic flow discovery algorithm

Given the discrete loss $\hat{L}_{m,n}$, the model-recovery algorithm is based on computing predicted snapshots corresponding to a candidate estimate $\hat{f}$ and comparing them to the observed ones. The predictions will be determined by solving the obtained ODE, i.e. equation 1 where $f$ is replaced by $\hat{f}$, through a numerical integration scheme, such as Runge–Kutta solvers. In parallel, the optimization over candidate estimates is

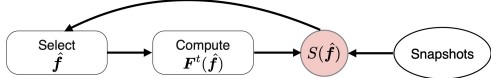

Figure 1: Building blocks of the proposed algorithm

realized using a MCTS-based symbolic search. For that matter, the goal of MCTS will be to maximize a score function defined for a differentiable $\hat{f} : \mathbb{R}^d \longrightarrow \mathbb{R}^d$ by

$$V(\hat{f}) = \frac{1}{1 + \hat{L}_{m,n}(\hat{f})}.$$

To ensure permutation invariance in the obtained expressions, which is characteristic of network systems, we restrict the search space by applying a permutation invariant aggregation operation (e.g. *sum*) to each expression before each evaluation. That is, we restrict our search to permutation-invariant expressions by design. We report an ablation study corresponding to the removal of the permutation invariance module in subsection 5.2. The pseudo-code of the algorithm is summarized in *Algorithm* 1, where the main steps are illustrated in Fig. 1, with more details in Appendix D. As for the implementation of $\hat{L}_{m,n}$, it reduces when comparing two empirical distributions $\hat{\mu} = \frac{1}{m} \sum_{i=1}^{m} \delta_{\alpha_i}$ and $\hat{\nu} = \frac{1}{m} \sum_{j=1}^{m} \delta_{\beta_j}$ to solving the following linear program

$$\min_{P \in U(\hat{\mu},\hat{\nu})} \langle P, M \rangle$$

where $M = (\|\alpha_i - \beta_j\|_2^2)_{i,j}$ is the cost matrix and $U$ the set of couplings between $\hat{\mu}$ and $\hat{\nu}$. Hence, its computational cost is in $O(dm^2 + m^3)$, where $m$ is the number of samples and $d$, the dimension of the system.

**Remark 4.1**
The restriction of the setup to vector fields $f$ such that the ODE system $\dot{y} = f(y)$ has a well-defined solution, is implicitly made by restricting our attention to states $t \mapsto x_t$ defined on $[0, T]$. From a practical perspective, if a field $f$ leading to a solution that explodes before $T > 0$ is selected, then such a field will be automatically discarded by the search algorithm, since its loss function will be very large.

**Remark 4.2**
H represents here the equivalent of number of epochs in training a deep learning model. As for N, it represents the number of runs, equivalent to training a neural network for different seeds and keeping the best model. Hence, the larger these parameters the more likely a better performance is obtained. As for the potential for over-fitting that is encountered in training a neural network for too long, it does not occur here because the optimization is discrete, hence the best performing model is saved anyway.

---

**Algorithm 1** Symbolic Distribution Flow Learner

---

**Inputs:** Number of episodes $N$, number of roll-outs $H$, screen-shots $(\hat{\mu}_{t_i,m})_{i,m}$ at $(t_i)_{0 \leq i \leq n}$
**Initialization:** Estimate the value of each operation $(+, -, \times, \sin, \dots)$ as a root node ;
**for** $e = 1, \dots, N$ **do**
    Randomly select a root node and build an expression tree as follows:
    **if** Tree is complete **then**
        Evaluate the corresponding estimate $\hat{f}$ by computing $V(\hat{f})$, Back-propagate the obtained value
    **else**
        Run $H$ roll-outs, Store the best estimate, Back-propagate the corresponding value
        Select the operation $a$ maximizing $UCT(s, a)$ where $s$ is the current state of the tree
    **end if**
**end for**
**Return:** Most accurate $\hat{f}$ over the $N$ episodes

---

### 4.3 Monte-Carlo Tree Search sample complexity

A crucial question in MCTS-based algorithms is to estimate the number of required episodes, executed by the algorithm, in order to honor a certain error tolerance in the obtained solution of the target optimization problem. We apply a non-asymptotic error analysis result by (Shah et al., 2020) to determine the minimal number of episodes -also known as sample complexity- that ought to be used.

**Proposition 1.**
The average number of score evaluations $E$ required for the MCTS algorithm[2] to find an $\varepsilon$-optimal[3] solution where $\varepsilon > 0$, is at most given by:

$$E = O\left(q \cdot \varepsilon^{-(4+M)} \cdot (\log \frac{1}{\varepsilon})^5\right) \ ,$$

where $M$ is the maximum allowed expression length and $q$ the size of the chosen elementary function set.

*Proof.* The proof is postponed to Appendix B. □

## 5 Numerical Experiments

First, we illustrate the performance of the proposed algorithm on the Kuramoto network system of ODEs, used across the biological, chemical and electrical domains to model circadian oscillators, pacemaker cells in the heart and electrical power networks among other applications (Discacciati & Hesthaven, 2021; Dörfler & Bullo, 2014). Then, we conduct an evaluation on a real-world dataset of embryoid stem cell trajectories (Moon et al., 2019). We provide comparisons of our algorithm with the current trajectory inference state-of-the-art algorithms, namely `TrajectoryNet` (Tong et al., 2020) and `JKOnet` (Bunne et al., 2022). `TrajectoryNet` relies on a (neural net-based) continuous normalizing flow (Grathwohl et al., 2018) formulation, augmented with relevant regularizations such as growth rate and velocity penalization. `JKOnet` builds upon the celebrated JKO scheme (Jordan et al., 1998) describing energy gradient flows, where it parameterizes the energy function and the Monge potential using Input Convex Neural Networks (Amos et al., 2017). In addition to being based on black-box models -in this instance, neural nets- these approaches contrast with `SDFL` in that they require (extra)-hyper-parameter tuning.

For the experimental comparison, we retrain the models with the architectures and hyper-parameters proposed by the respective authors (Tong et al., 2020; Bunne et al., 2022); however, we employ early-stopping to avoid over-fitting to the smaller data-sets. For `JKOnet`, we use a small regularization parameter $\varepsilon = 0.001$ to make its target closer to the Wasserstein distance, which is our evaluation metric. We conduct experiments in the small and noisy data regime with varying Training Sample Size (TSS) per snapshot, illustrating the

---

[2]With UCT defined as in (Shah et al., 2020).
[3]An $\varepsilon$-optimal solution of an optimzation problem $\min_{x \in E} g(x)$ is a value $x_\varepsilon \in E$ that satisfies $g(x_\varepsilon) \leq \min_{x \in E} g(x) + \varepsilon$.

suitability of `SDFL` for practical costly data-collection conditions. Similar to previous studies, numerical evaluations are conducted based on the Wasserstein distance between the predicted empirical distributions and the ground-truth distributions (averaged over 3 runs) on unseen data. The computational time of the different methods is reported in section 5.3. For completeness, we report a performance comparison of `SDFL` against the symbolic regression state-of-the-art methods by Qian et al. (2022) and Sun et al. (2023) in Appendix E.4, although they represent less challenging competitors.

## 5.1 Kuramoto system of ODEs

We investigate the recovery of the Kuramoto system with the state variable $x_t = (\theta_i(t))_{1 \leq i \leq d}$ following

$$\dot{\theta}_i(t) = \omega_i + \frac{1}{d} \sum_{j=1}^{d} K_{ij} \sin\left(\theta_j(t) - \theta_i(t)\right),$$

where $(\omega_i)_{1 \leq i \leq d}$ are the corresponding natural frequencies. Moreover $\mathcal{G} = (K_{ij})_{1 \leq i,j \leq d}$ is the graph weight matrix. Gaussian initial condition (with mean 2 and unity variance) is employed. For simplicity, following (Discacciati & Hesthaven, 2021), we assume a fully connected uniformly weighted graph (i.e. we take $K_{ij} = K$ for all $i, j \in \{1, \ldots, d\}$). We consider $n = 15$ snapshots, in $d = 3$ dimensions, over the time horizon $T = 30$. Furthermore the natural frequencies are set to 0.01. To highlight the accuracy and robustness of our approach, Figure 2 illustrates the inferred trajectory of state variable (using `SDFL`), and Figure 3 the robustness of Wasserstein distance with respect to the noise. In the latter figure, the distances are computed after training. Note that to reduce the `SDFL` running time, it was run in two steps where the global function structure is

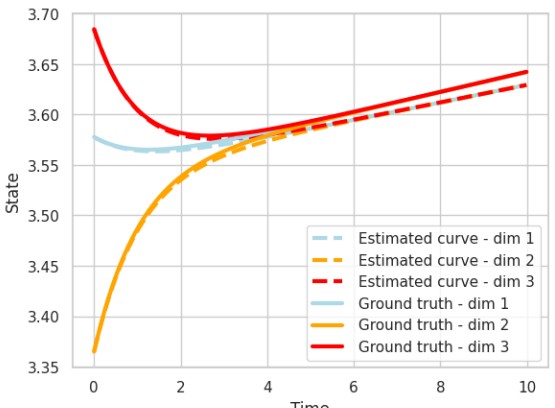

Figure 2: Inferred and ground truth state trajectory

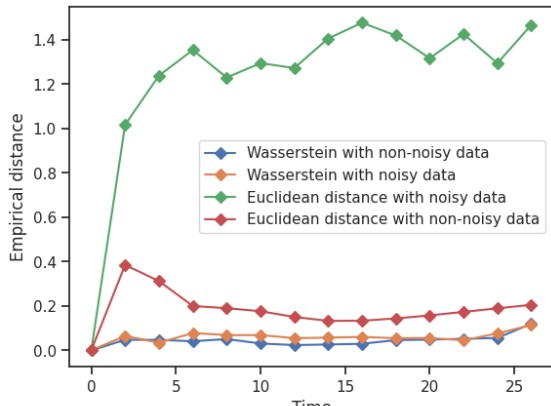

Figure 3: Distance between inferred and reference distributions, by different metrics

fitted first, followed by the estimation of regression coefficients (see Appendix C for details). The results reported in table 1 (representing the test error for a sample size of $m = 50$) suggest `SDFL` is competitive with the state-of-the-art. The system of equations obtained by `SDFL` with 150 samples per screen-shot is given by

Table 1: Prediction error comparison in the Wasserstein metric for the Kuramoto model

| TSS | SDFL | JKONet | TrajectoryNet |
|---|---|---|---|
| 50 | $0.57 \pm 0.04$ | $0.86 \pm 0.22$ | $4.33 \pm 0.13$ |
| 100 | $0.57 \pm 0.06$ | $0.76 \pm 0.18$ | $4.83 \pm 0.10$ |
| 150 | $0.46 \pm 0.05$ | $0.76 \pm 0.22$ | $3.15 \pm 0.11$ |

$$\begin{cases} \dot{\theta}_1 = 0.0087 + 0.3293 * (\sin(\theta_2 - \theta_1) + \sin(\theta_3 - \theta_1)) \\ \dot{\theta}_2 = 0.0087 + 0.3293 * (\sin(\theta_1 - \theta_2) + \sin(\theta_3 - \theta_2)) \\ \dot{\theta}_3 = 0.0087 + 0.3293 * (\sin(\theta_2 - \theta_3) + \sin(\theta_1 - \theta_3)) \end{cases}$$

Additionally, we illustrate the predicted distributions by `SDFL` against the ground truth for snapshot time $t = 2$ for two different dimensions, with Gaussian mixture as initial condition. Figure 4 translates the Wasserstein prediction errors to a more intuitive and visual comparison.

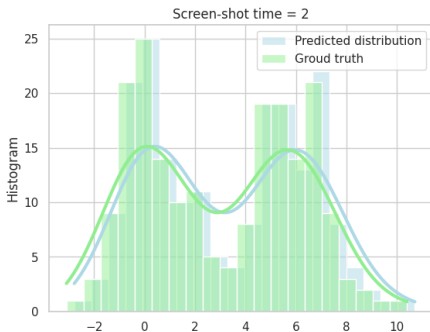

Figure 4: Un-normalized empirical density comparison for the $1^{st}$ dimension

## 5.2   Single-Cell Population Dynamics

We consider the problem of learning the evolution of embryonic stem cells based on single-cell RNA sequencing data measurements over a period of 27 days, where the data was collected at 5 different snapshots (see Moon et al. (2019)). Apart from the high cost of obtaining a large dataset, a key difficulty in this setting is that cells are (usually) destroyed during measurement (Bunne et al., 2022). Hence, there is a need for schemes which predict the distribution evolution across time, rather than individual trajectories, from limited observations. The dataset was pre-processed using the recent dimensionality reduction technique PHATE (Moon et al., 2019), which is specifically designed to preserve maximum variablity in the low-dimensional space while allowing for intuitive visualization, and therefore interpretation. With this goal in mind, a natural choice for the reduced dimension is $d = 3$.

Table 2: Prediction error comparison in the Wasserstein metric for embryoid stem cell trajectory inference

| TSS | SDFL | JKONet | TrajectoryNet |
|-----|------|--------|---------------|
| 100 | $0.97 \pm 0.14$ | $3.18 \pm 0.54$ | $6.78 \pm 0.26$ |
| 200 | $0.87 \pm 0.11$ | $3.72 \pm 0.51$ | $6.65 \pm 0.06$ |
| 300 | $0.79 \pm 0.16$ | $3.03 \pm 0.70$ | $6.25 \pm 0.11$ |

Similar to the previous case of Kuramoto network, `SDFL` outperforms the competitor state-of-the-art methods, as shown in table 2 with a test sample size $m = 50$. This is encouraging, specifically since for the cell population dynamics, the dataset is not produced based on an underlying mathematical model. Most importantly, it confirms the relevance of `SDFL` as a dynamic regression tool for real-world data subject to noise and limited observability. It should be emphasized though, that neural net based approaches (`TrajectoryNet` and `JKONet`) require sufficiently large amounts of data to reach their optimal performance. As for `JKONet`, one explanation for why it performs better than `TrajectoryNet`, is that it enforces an inductive bias through the JKO scheme (Bunne et al., 2022). We note that the performance of the `SDFL` model is stable across

training sample sizes. The system of ODEs obtained by `SDFL` with $m = 50$ samples per snapshot is given by

$$\begin{cases} \dot{x_1} = \cos(x_2) * x_1 + \cos(x_3) * x_1 \\ \dot{x_2} = \cos(x_1) * x_2 + \cos(x_3) * x_2 \\ \dot{x_3} = \cos(x_2) * x_3 + \cos(x_1) * x_3 \end{cases}$$

It is worth noting that network modeling is most relevant for scRNAseq trajectory inference, as extensively demonstrated in static settings (Wang et al., 2021; Van Dijk et al., 2018; Van de Sande et al., 2020). For `SDFL`, we demonstrate the importance of enforcing permutation invariance through a simple ablation study. More precisely, we report in table 3 below the performance metric of the scheme with and Without the Permutation Invariance (WPI) module, for the scRNA-seq evolution modeling task, without fitted regression parameters. We observe a drop in performance given the same search horizon (number of episodes).

Table 3: Prediction loss for the scRNA-seq evolution modeling task

| Method | SDFL | SDFL-WPI |
|---|---|---|
| $\mathcal{W}_2$ metric | 2.0822 | 2.3247 |

### 5.3 Computational time

We present in tables 4 a computational time comparison between `SDFL`, `JKOnet` and `TrajectoryNet`. For a fair comparison, all the reported running times are obtained on an Intel(R) Core(TM) i7-7500U CPU. The reported times correspond to training on a sample of $m = 50$ per screen-shot for the Kuramoto

Table 4: Average running time (hours) for the Kuramoto system and the scRNA-seq evolution modeling

| Task | SDFL | JKONet | TrajectoryNet |
|---|---|---|---|
| Kuramoto | 5.8166 | 0.7044 | 5.7709 |
| scRNA-seq | 3.8777 | 0.4696 | 3.8472 |

system modeling task and $m = 100$ per screen-shot for the scRNA-seq evolution modeling one. It is worth emphasizing that the reported running times for `JKONet` and `TrajectoryNet` do not take into account hyper-parameter tuning. And, since `SDFL` does not require the latter, we believe it is still competitive (for low dimensions).

**Remark 5.1**
It is worth noting that, even for low-dimensional settings, extracting explicit models under the sparse data regime (one measured distribution every few days) could not be achieved manually.

**Remark 5.2**
To give an estimate of the corresponding computational time for higher dimensional systems, considering the same expression length for all dimensions, note that the computational time is of the order of $O(q^d)$, where $q$ is the size of the set of elementary functions. Hence, if we consider a system of dimension 5 instead of 3, the computational time of SDFL would approximately be multiplied by 50, with the chosen elementary function set. We further discuss this limitation in section 6 below. As for opportunities for optimization, there are at least two of them: one is to parallelize the stochastic roll-outs and the other is to parallelize the runs of different episodes. This can be done easily given the structure of the scheme.

# 6 Explainability & Limitations

Given that the models obtained by SDFL are based on compact symbolic expressions, they allow for the extraction of interpretable insights (Qian et al., 2022; Menara et al., 2022). For instance, the obtained model for the single-cell population dynamics suggests complex oscillating patterns that are not captured by a mere linear interaction between the components, as illustrated by the plot comparisons in figure 5 below[4] and the accuracy comparison in table 5 (Appendix E). Additionally, such explicit models can allow domain experts to conduct sensitivity and parametric analyses, revealing critical parameters that result in jumps or significant changes in the system. One specific example of such behaviors is the critical coupling strength of the Kuramoto model (Discacciati & Hesthaven, 2021). Specifically, if $K = K_{i,j} < \frac{4}{\pi}$ then

$$R = \left| \frac{1}{d} \sum_{j=1}^{d} e^{i\theta_j(T)} \right| = O(1/\sqrt{d})$$

denoting incoherent behavior. On the other hand, if $K \gg \frac{4}{\pi}$ then $R \simeq 1$, denoting complete synchronization. However, such advantages come at the cost of challenging scalability to higher dimensions (Virgolin & Pissis, 2022), as well as the need to solve candidate ODEs on the fly which can be computationally intensive.

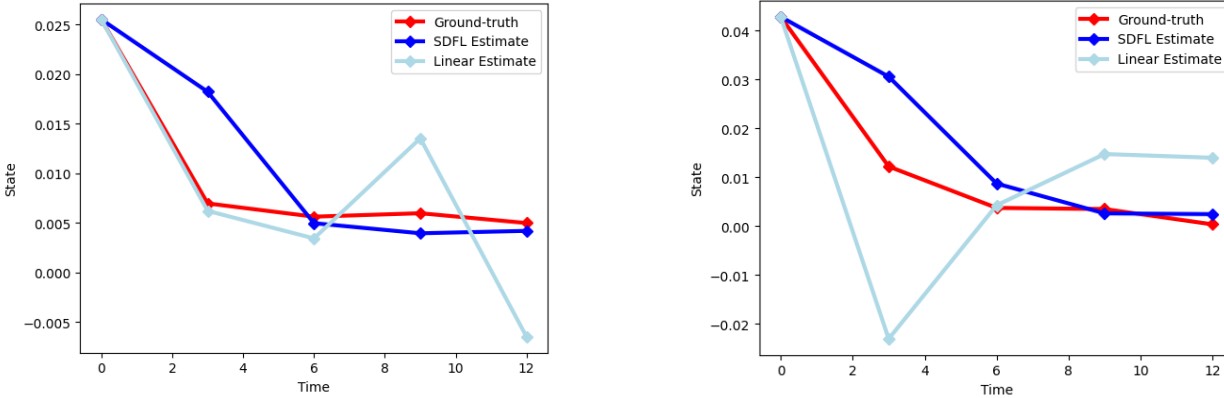

Figure 5: Inferred trajectories for different samples

# 7 Conclusion

In this work, we investigated model-discovery of dynamic systems, subject to randomness. Specifically, taking a symbolic regression perspective, we considered a setup in which only a few random observations of the system are provided in time and space. We proposed SDFL, a scheme incorporating several innovative ideas, in order to tackle this multifaceted dynamic inference problem. In particular, we introduced an appropriate measure for goodness-of-fit by devising a time integrated Wasserstein loss. This design choice turned out to be powerful by offering robustness of the model-discovery in presence of noise/uncertainty in the data. In addition, we derived theoretical guarantees on the scheme sample complexity. Motivated by the performance of SDFL in our numerical experiments, we believe these developments contribute to robust model-discovery approaches from noisy and limited data, and bridge the rich literatures on probabilistic modeling and symbolic regression. Computational scaling to higher dimensions remains an interesting open problem for future studies.

---

[4]Note that, the estimates are based on normalized data.

## Acknowledgments

This work was supported by the Swiss National Science Foundation under grant No. 212876. We acknowledge computational resources from the Swiss National Supercomputing Centre CSCS.

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

## A  Proof of Theorem 1

*Proof.* Recall that we would like to prove that there are positive constants $M_1, M_2 > 0$ such that:

$$\mathbb{E}\left|\frac{1}{n}\sum_{i=1}^{n}\mathcal{W}_2(F^{t_i}(\hat{f})_{\#}\hat{\mu}_{t_0,m}, \hat{\mu}_{t_i,m}) - \frac{1}{T}\int_0^T \mathcal{W}_2(F^t(\hat{f})_{\#}\mu_0, \mu_t)\,\mathrm{d}t\right| \leq \frac{M_1}{m^{(d/2+1)}} + \frac{M_2}{n}$$

provided $m, n \geq 2$. Without loss of generality, we assume $T = 1$. By theorem 1 in (Fournier & Guillin, 2015), given that $x_0 \mapsto F^t(x_0)$ is continuous and for all $i \in \{1, \dots, n\}$, $\mu_{t_i}$ has compact support on the Polish space $\mathbb{R}^d$, there exists $C_1, C_2 > 0$ such that

$$\mathbb{E}\left[\frac{1}{n}\sum_{i=1}^{n}\mathcal{W}_2(F^{t_i}(\hat{f})_{\#}\hat{\mu}_{t_0,m}, F^{t_i}(\hat{f})_{\#}\mu_{t_0})\right] \leq \frac{C_1}{m^{(d/2+1)}}$$

$$\text{and} \qquad \mathbb{E}\left[\frac{1}{n}\sum_{i=1}^{n}\mathcal{W}_2(\hat{\mu}_{t_i,m}, \mu_{t_i})\right] \leq \frac{C_2}{m^{(d/2+1)}} \ .$$

Therefore, thanks to the triangular inequality satisfied by $\mathcal{W}_2$, we get

$$\mathbb{E}\left|\frac{1}{n}\sum_{i=1}^{n}\mathcal{W}_2(F^{t_i}(\hat{f})_{\#}\hat{\mu}_{t_0,m}, \hat{\mu}_{t_i,m}) - \frac{1}{n}\sum_{i=1}^{n}\mathcal{W}_2(F^{t_i}(\hat{f})_{\#}\mu_{t_0}, \mu_{t_i})\right| \leq$$

$$\mathbb{E}\left[\frac{1}{n}\sum_{i=1}^{n}\mathcal{W}_2(F^{t_i}(\hat{f})_{\#}\hat{\mu}_{t_0,m}, F^{t_i}(\hat{f})_{\#}\mu_{t_0})\right] + \mathbb{E}\left[\frac{1}{n}\sum_{i=1}^{n}\mathcal{W}_2(\hat{\mu}_{t_i,m}, \mu_{t_i})\right] \leq \frac{C_1 + C_2}{m^{(d/2+1)}} \ . \qquad (5)$$

On the other hand, denoting by $\mathcal{M}(\mathbb{R}^d \times \mathbb{R}^d)$ the Banach space of finite measures on $\mathbb{R}^d \times \mathbb{R}^d$ equipped with the total variation norm, the mapping $(t, \gamma) \in [0,1] \times \mathcal{M}(\mathbb{R}^d \times \mathbb{R}^d) \mapsto \int_{\mathbb{R}^d \times \mathbb{R}^d} \|F^t(\hat{f})(x) - y\|_2^2\,\mathrm{d}\gamma(x,y)$ is differentiable. Moreover, by compactness of the set of couplings given two fixed marginals and by theorem 5.20 in (Villani, 2009), conditions $(ii)$ and $(iii)$ of theorem 4.24 in (Bonnans & Shapiro, 2013) are satisfied leading to the fact that the mapping $t \mapsto \mathcal{W}_2(F^t(\hat{f})_{\#}\mu_0, \mu_t)$ is continuously differentiable on $[0,1]$[5]. Hence, there exists a positive constant $C_3 > 0$ such that

$$\left|\int_{(i-1)/n}^{i/n} \mathcal{W}_2(F^t(\hat{f})_{\#}\mu_0, \mu_t)\,\mathrm{d}t - \frac{1}{n}\mathcal{W}_2(F^{t_i}(\hat{f})_{\#}\mu_{t_0}, \mu_{t_i})\right| \leq \frac{C_3}{n^2}$$

for $i \in \{1, ..., n\}$, which yields

$$\left|\int_0^1 \mathcal{W}_2(F^t(\hat{f})_{\#}\mu_0, \mu_t)\,\mathrm{d}t - \frac{1}{n}\sum_{i=1}^{n}\mathcal{W}_2(F^{t_i}(\hat{f})_{\#}\mu_{t_0}, \mu_{t_i})\right| \leq \frac{C_3}{n} \ .$$

Consequently, by summation with (5), we get

$$\mathbb{E}\left|\frac{1}{n}\sum_{i=1}^{n}\mathcal{W}_2(F^{t_i}(\hat{f})_{\#}\hat{\mu}_{t_0,m}, \hat{\mu}_{t_i,m}) - \int_0^1 \mathcal{W}_2(F^t(\hat{f})_{\#}\mu_0, \mu_t)\,\mathrm{d}t\right| \leq \frac{C_1 + C_2}{m^{(d/2+1)}} + \frac{C_3}{n} \ .$$

$\square$

## B  Proof of Proposition 1

*Proof.* Denote by $(\mu_t)_{t \in [0,T]}$ the ground-truth probability flow and let $f$, $g$ be the continuously differentiable functions defined by two analytic expressions from the selection space of MCTS. Recall that the goal is to maximize the functional given by

$$V : f \mapsto \frac{1}{1 + \hat{L}_{m,n}(f)} \ .$$

---

[5]Given that the optimal coupling is unique and that the directional derivative is continuous with respect to $t \in [0, 1]$ -as a supremum of a differentiable parametric family of convex functions.

The result is obtained by showing that the conditions of theorem 2 in (Shah et al., 2020) are satisfied in our setting. For that matter, it suffices[6] to show that

$$L : f \mapsto \int_0^T \mathcal{W}_2(F^t(f)_{\#}\mu_0,\, \mu_t) \, \mathrm{d}t$$

is Lipschitz with respect to the $L^1$ or $\| \cdot \|_\infty$ norms; since the derivative of $h : x \mapsto \frac{1}{1+x}$ is bounded on $\mathbb{R}^+$ implying that $h$ is Lipschitz. Furthermore, we have

$$|S(f) - S(g)| \leq \int_0^T \left| \mathcal{W}_2(F^t(f)_{\#}\mu_0,\, \mu_t) - \mathcal{W}_2(F^t(g)_{\#}\mu_0,\, \mu_t) \right| \, \mathrm{d}t$$

$$\leq \int_0^T \mathcal{W}_2(F^t(f)_{\#}\mu_0, F^t(g)_{\#}\mu_0) \, \mathrm{d}t$$

because $\mathcal{W}_2$ is a distance. Additionally, since both distributions have compact support, and by regularity of $F^t(f), F^t(g)$, there exists constants $C, C' > 0$ such that for all $t \in [0, T]$,

$$\mathcal{W}_2(F^t(f)_{\#}\mu_0, F^t(g)_{\#}\mu_0) \leq C \cdot \mathcal{W}_1(F^t(f)_{\#}\mu_0, F^t(g)_{\#}\mu_0)$$

$$\leq C \cdot \sup_{\mathrm{Lip}(q) \leq 1} \left| \int q \, \mathrm{d}(F^t(f)_{\#}\mu_0) - \int q \, \mathrm{d}(F^t(g)_{\#}\mu_0) \right|$$

$$\leq C \cdot \sup_{\mathrm{Lip}(q) \leq 1} \left| \int q \circ F^t(f) \, \mathrm{d}\mu_0 - \int q \circ F^t(g) \, \mathrm{d}\mu_0 \right|$$

$$\leq C \cdot \sup_{\mathrm{Lip}(q) \leq 1} \int \left| q \circ F^t(f) - q \circ F^t(g) \right| \, \mathrm{d}\mu_0$$

$$\leq C \cdot \int \left| F^t(f) - F^t(g) \right| \, \mathrm{d}\mu_0$$

$$\leq T \cdot C' \cdot \| f - g \|_\infty$$

where the second inequality is justified by the dual representation of $\mathcal{W}_1$, the third by the change of variable formula[7], and the fifth by the Lipschitz property of $q$. The last inequality is justified by the fact that the solutions $t \mapsto F^t(f)(x)$ and $t \mapsto F^t(g)(x)$ of the differential equations $\dot{y} = f(y)$ and $\dot{y} = g(y)$ respectively, for a given initial condition $x \in \mathbb{R}^d$, are fixed points of the Picard operator[8], which is Lipschitz.

Consequently, integrating over $t \in [0, T]$ on both sides, we obtain the Lipschitz property of $f \mapsto \frac{1}{1+L(f)}$. Finally, since $L(f)$ can be approximated by $\hat{L}_{m,n}(f)$ with arbitrary accuracy, we get the sample complexity upper-bound. $\qquad \square$

## C   Implementation details & Reproducibility

For the implementation of SDFL, we set the building operations consisting of $\{+, -, \times, \div, \cos, \sin, \exp\}$ with a maximum of $L = 20$ operations per expression, with a number of episodes of 500 to 1000. For the recovery of the Kuramoto system, we use 15 snapshots with time-stamps $t_i = 2i$ for $1 \leq i \leq 15$, and we set $K = 1/3$. The fitted parameters correspond to $m = 50$ sample points per screenshot.

For the cellular dynamics data, after a pre-processing step using PHATE (Moon et al., 2019), the dimension is reduced to $d = 3$ then standard Gaussian noise samples were added to the training set. The following governing ODE system is obtained (after 50 episodes):

$$\begin{cases} \dot{x_1} = \cos(x_2) * x_1 + \cos(x_3) * x_1 \\ \dot{x_2} = \cos(x_1) * x_2 + \cos(x_3) * x_2 \\ \dot{x_3} = \cos(x_2) * x_3 + \cos(x_1) * x_3 \end{cases} \qquad (6)$$

---

[6]Note that, since the considered state space is finite, we do not need to construct an explicit covering.

[7]Note that $q \circ F^t(f)$ denotes the composition of $q$ by $F^t(f)$.

[8]The Picard operator is introduced in the proof of the existence of (local) solutions to ODEs (see e.g. Coddington et al. (1956)).

Given the inductive bias of searching for an explicit ODE model, we note that `SDFL` results in a model with similar accuracy across sample sizes (of the same order, as those illustrated in table 2). For comparison, we used the publicly available implementations of `JKOnet` and `TrajectoryNet`, from (Bunne et al., 2022) and (Tong et al., 2020) respectively. we retrain the competitor models with the architectures and hyper-parameters proposed by the respective authors (Tong et al., 2020; Bunne et al., 2022); however, we employ early-stopping to avoid over-fitting to the smaller data-sets[9]. For `JKOnet`, we use a small regularization parameter $\varepsilon = 0.001$ to make its target closer to the Wasserstein distance. Additionally, to foster reproducibility, a Python implementation of `SDFL` has been made public at `https://github.com/Ramzisofo/SDFL`.

## D  Detailed pseudo-code

Below, we provide a more detailed pseudo-code of `SDFL` while featuring more clearly the MCTS components.

---

**Algorithm 2**  Symbolic Distribution Flow Learner [extended description]

---
1: **Inputs:** Number of episodes $N$, number of roll-outs $H$, maximal expression length $M$, elementary functions set $(+, -, \times, \sin, \dots)$, screen-shots $(\hat{\mu}_{t_i, m})_{i,m}$ at $(t_i)_{0 \le i \le n}$
2:
3: **Initialization:**
4:  $\to$ Estimate the value of each operation $(+, -, \times, \sin, \dots)$ as a root node through $H$ stochastic roll-outs
5:  $\to$ Store these values in $V(0, a)$, where $s = 0$ represents the empty tree state and $a$ the chosen root operation
6:  $\to$ Define $S_{\max} := \max_a V(0, a)$
7:
8: **for** $e = 1, \dots, N$ **do**
9:   Randomly select a root node and build an expression tree as follows:
10:   **if** Tree is complete **then**
11:     Evaluate the corresponding estimate $\hat{f}$ by computing $S(\hat{f})$
12:     **if** $S(\hat{f}) > S_{\max}$ **then**
13:       $S_{\max} := S(\hat{f})$
14:       Back-propagate the obtained value $S(\hat{f})$ by updating the values of $(V(s_p, a))_{p \ge 1}$ where $(s_p)_{p \ge 1} = [a_0, a_1, \dots, a_{p-1}]$ is the finite sequence of encountered tree states before completion
15:     **end if**
16:   **else**
17:     Run $H$ roll-outs by randomly selecting operations to extend and complete the tree
18:     Assign a value of 0 to trees resulting in an inconsistent mathematical expression
19:     Store the best estimate in $V(s, a_b)$ where $a_b$ is the corresponding best operation
20:     Back-propagate the obtained value of $V(s, a_b)$ to the encountered tree states $(s_p)_{p \ge 1} = [a_0, a_1, \dots, a_{p-1}]$
21:     Select the operation $a$ maximizing $UCT(s, a)$ where $s$ is the current state of the tree
22:   **end if**
23: **end for**
24: **Return:** Most accurate $\hat{f}$ over the $N$ episodes

---

[9]Number of iterations used were 1000 for `JKOnet` and 1500 for `TrajectoryNet`

**Remarks:**

- The back-propagation, such as in line 14, consists in updating the values of the tree states that have been encountered until that step, in that specific episode. Note that the tree states consist of the sequence of operations which have been selected to constitute the chosen expression (until that step).

- The idea is to keep track of the operations (nodes) leading to a good score, and to give them higher chance of being selected in the next rounds.

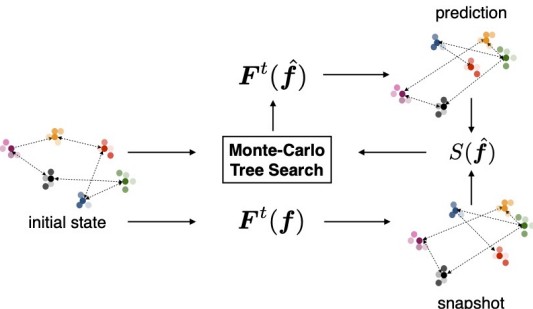

Figure 6: The devised model-discovery setup.

## E  Additional numerical results

### E.1  Robustness in higher dimensions

We demonstrate numerically the robustness of the the proposed loss function for higher dimensions. As shown in Figures 7, 8 and 9, the robustness property extends to higher dimensions.

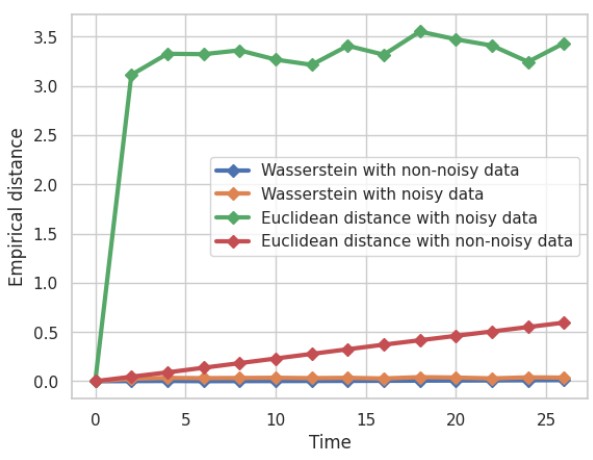

Figure 7: Distance between inferred and reference distributions, for system dimension $d = 5$

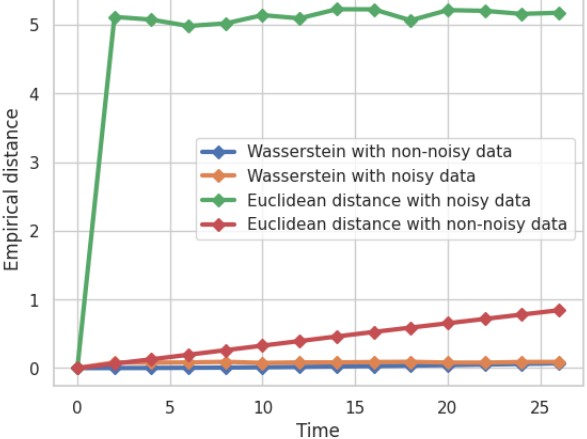

Figure 8: Distance between inferred and reference distributions, for system dimension $d = 10$

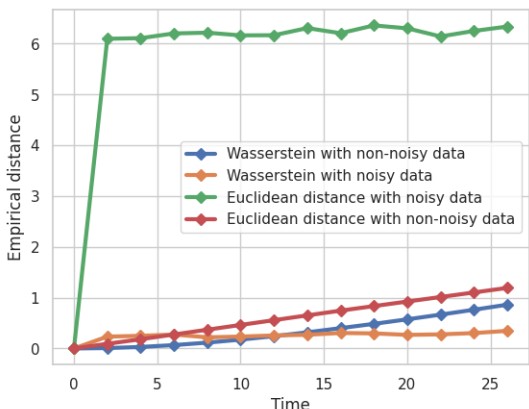

Figure 9: Distance between inferred and reference distributions, for system dimension $d = 20$

### E.2 Heterogeneous Kuramoto system

We run the proposed scheme on data obtained from 50 trajectories, measure at time points $t_0 = 0$, and $t_{i+1} = 1 + t_i$, for $1 \leq i \leq 15$, for the Kuramoto system given by

$$\begin{cases} \dot{\theta}_1 = 0.01 + 1.333 * (\sin(\theta_2 - \theta_1) + \sin(\theta_3 - \theta_1)) \\ \dot{\theta}_2 = 0.01 + 1.222 * (\sin(\theta_1 - \theta_2) + \sin(\theta_3 - \theta_2)) \\ \dot{\theta}_3 = 0.01 + 1.111 * (\sin(\theta_2 - \theta_3) + \sin(\theta_1 - \theta_3)) \end{cases}$$

We note that the scheme recovers the global structure of the system as expected. As for the parameters, given the limited amount of data and non-convexity of the estimation, the true parameters have not been successfully recovered.

### E.3 Comparison against symbolic baselines

We provide a numerical comparison of the performance of the model obtained by `SDFL` in comparison with (symbolic) baselines, over 3 runs following (Tong et al., 2020; Bunne et al., 2022). As illustrated in table 5, we note that the model obtained by `SDFL` offers a significant improvement, suggesting its usefulness for domain experts, with an interest for explicit representations of dynamic processes.

Table 5: Prediction loss for the scRNA-seq evolution modeling task

| Model | Error |
|---|---|
| Obtained by `SDFL` | $2.02 \pm 0.34$ |
| Linear | $3.99 \pm 1.23$ |
| Independent components | $3.41 \pm 0.76$ |

Table 6 displays the test error for the models, where linear regression parameters were estimated based on the general symbolic structure obtained previously. The results suggest once again that the model obtained by `SDFL` captures the non-linear aspect of the system behavior.

### E.4 Comparison against symbolic regression state-of-the-art

We compare `SDFL` estimates against those obtained by the state-of-the-art methods `SPL` (Sun et al., 2023) and `D-Code` (Qian et al., 2022) for varying sampling frequencies, using the publicaly available code by the

Table 6: Prediction loss for the scRNA-seq evolution modeling task, with additional regression parameters

| Model | Error |
|---|---|
| Obtained by `SDFL` | $0.94 \pm 0.41$ |
| Linear | $1.28 \pm 0.55$ |
| Independent components | $3.41 \pm 0.76$ |

respective authors[10]. The point of this section is to show numerically that, under similar sample sizes, `SDFL` needs much fewer data-points across time than the current state-of-the-art. We use the Kuramoto model as a benchmark and note that it would not be possible to run the same comparisons on the scRNAseq dataset, since in that case we do not have the possibility to change the sampling frequency. We compare the `SDFL` estimate with a sampling frequency of $freq = 0.5$, that is corresponding to $t_{i+1} - t_i = 2$ where $t_i$ are the screenshot times, against `SPL` and `D-Code` estimates with a sampling frequency of $freq = 100$, that is $t_{i+1} - t_i = 0.01$. We report in tables 7 and 8 the prediction loss, where the Kuramoto system has been solved with a maximum integration step of 0.1 and 0.01 respectively. Additionally, we report in table 9 below the accuracy evolution across sampling frequencies of the competitor methods. We note that `SDFL` requires 100 times less sampling frequency to reach comparable accuracy.

Table 7: Normalized prediction loss for the Kuramoto system, with integration step of 0.1

| Method | Wasserstein Error |
|---|---|
| `SDFL` | $0.34 \pm 0.24$ |
| `SPL` | $2.54 \pm 0.62$ |
| `D-Code` | $0.34 \pm 0.05$ |

Table 8: Prediction loss for the Kuramoto system, with integration step of 0.01

| Method | Wasserstein Error |
|---|---|
| `SDFL` | $1.39 \pm 0.59$ |
| `SPL` | $61.13 \pm 5.47$ |
| `D-Code` | $188.48 \pm 52.74$ |

Table 9: Normalized prediction loss evolution across sampling frequencies

| Frequency | 100 | 10 | 1 |
|---|---|---|---|
| `SPL` | $2.54 \pm 0.62$ | $7.45 \pm 3.57$ | $10.36 \pm 1.00$ |
| `D-Code` | $0.34 \pm 0.05$ | $6.53 \pm 0.92$ | $23.72 \pm 2.66$ |

We observe that the accuracy of the different methods increases with the sampling frequency as expected. Note that this remark goes in the same direction as the findings of the papers Sun et al. (2023) and Qian et al. (2022), that proposed `SPL` and `D-Code` respectively, which limit their evaluations to $freq = 100$ (i.e. 100 points per unit of time) or notice a drop in performance for lower sampling frequencies. As for the important drop of performance in the case of a smaller integration step size, it could be explained by the fact that the

---

[10]We use the hyper-parameters optimized by the respective authors with a running horizon $H$ given by $H_{\texttt{SDFL}} = 200$, $H_{\texttt{SPL}} = 500$ and $H_{\texttt{D-Code}} = 1500$.

estimates only recovered the global behavior of the Kuramoto system state and trying to match them to the true state at a fine scale leads to error compounding.

