# OpenReview forum: "Robust Symbolic Regression for Dynamical System Identification"
_TMLR — Accepted by TMLR_

### Review · Reviewer_mq2Y · 2024-11-07

**Summary Of Contributions:**

This work proposes a white box approach-dubbed Symbolic Distribution Flow Learner (SDFL). This approach leverages the symbolic search with a Wasserstein-based loss function. It also provides some theoretical guarantees of the model-discovery and complexity. Finally, it compares to the state-of-the-art on two dynamic systems.

**Audience:**

Yes

**Claims And Evidence:**

Yes

**Requested Changes:**

1. Some discussion on why authors apply MCTS-based algorithm to optimize and search for the trajectory? Discussion the benefits of symbolic search for optimization would be helpful.
2. Some ablation study for the metric selection would be interesting to explore as well.

**Strengths And Weaknesses:**

Strengths:
1. It proposes a novel trajectory inference approach based on symbolic search and Wasserstein distance.
2. Replacing neural network by the MCTS search is an interesting approach which takes better observability of the system

Weaknesses:
1. It is not clear how the inference is sensitive to some hyper-parameters in SDFL such as N and H.
2. It is a bit wired that the SOTA TrajectoryNet performs so worse on Kuramoto model and cell trajectory. What is the neural model size for those experiments? Some discussion may be useful.

---

> ### Author Response · Authors · 2025-02-07
>
> We thank the reviewer for the positive feedback and constructive comments, which we address below.
>
> It is not clear how the inference is sensitive to some hyper-parameters in SDFL such as N and H.
>
> $\longrightarrow$ We appreciate the reviewer's concern. H represents here the equivalent of number of epochs in  training a deep learning model. As for N, it represents the number of runs, equivalent to training a neural network for different seeds and keeping the best model. Hence, the larger these parameters the more likely a better performance is obtained. As for the potential for over-fitting that is encountered in training a neural network for too long, it does not occur here because the optimization is discrete, and we always retain the best performing model. We have added a remark clarifying this in the updated version.
>
> It is a bit wired that the SOTA TrajectoryNet performs so worse on Kuramoto model and cell trajectory. What is the neural model size for those experiments? Some discussion may be useful.
>
> $\longrightarrow$ Please note that we discuss this aspect in the second paragraph in the beginning of Section 5, and the second paragraph of Subsection 5.2. The hyper-parameters for the competitor models were those optimized by the respective authors. These models are already small (eg. two layer feedforward network with 64 neurons in each layer, in JKOnet). However, we employed early stopping to avoid over-fitting with smaller datasets. The reason that SDFL outperforms the neural network approaches is that the considered datasizes are small. This is aligned with the purpose of the study to tackle the small and noisy data regime, encountered in many applications, notably in biology.

---

> ### Author Response · Authors · 2025-02-07
>
> Requested Changes:
>
> Some discussion on why authors apply MCTS-based algorithm to optimize and search for the trajectory? Discussion the benefits of symbolic search for optimization would be helpful.
>
> $\longrightarrow$ We thank the reviewer for the comment. We have added more discussion on the advantages of MCTS in the first paragraph of Subsection 3.3
>
> Some ablation study for the metric selection would be interesting to explore as well.
>
> $\longrightarrow$ We thank the reviewer for the comment. Please note that the considered setting  (which is motivated by practical problems such as single-cell RNA sequence modeling) generally does not permit the direct computation of the Euclidean distance. This is because we observe discrete probability distributions which are not paired, i.e. they do not have the same support. As an example if we consider $\mu_1 =  \sum_{i=1}^n p_i \delta_{x_i}$ and $\mu_2 =  \sum_{j=1}^n q_j \delta_{y_j}$, we observe that the Euclidean distance between $\mu_1$ and $\mu_2$ can not be computed.In fact, even the KL divergence cannot be computed unless $\{ x_i, \, i \leq n\} = \{y_j, \, j \leq n  \}$. Additionally, ablating the loss would be equivalent to using MCTS as proposed by [1], which we compare against in Appendix E.3.
>
>
>
>
> [1] F. Sun, Y. Liu, J.-X. Wang, and H. Sun. Symbolic physics learner: Discovering governing
> equations via monte carlo tree search. In The Eleventh International Conference on
> Learning Representations, 2023

---

### Review · Reviewer_7KWH · 2024-12-11

**Summary Of Contributions:**

This paper introduces SDFL (Symbolic Distribution Flow Learner), a method for discovering interpretable ODEs that govern probabilistic flows in network systems from sparse/partial observations.

**Audience:**

Yes

**Claims And Evidence:**

Yes

**Requested Changes:**

As authors likely have anticipated, I have mainly questions about the runtimes/applicability beyond a proof of concept. I think the work satisfies the criteria for acceptance without these, and I did really appreciate the fact that authors were very direct about the shortcomings.

- Even though this is left as a future work, I would still like to see at least a simple example showcasing how the computational complexity would scale for higher dimensional systems? Even a simple back of the envelope calculation would be valuable.
- The runtime comparison suggests SDFL is significantly slower than baselines. Are there opportunities for optimization?

The paper presents a valuable contribution to the field of symbolic regression. Despite some potential limitations in scalability analysis and computational efficiency, the theoretical foundations are strong and the empirical results demonstrate clear advantages over existing methods. I believe the claims are well balanced and the topic is of interest, so I vote for acceptance.

**Strengths And Weaknesses:**

Strengths:

- Strong and convincing theoretical foundations
- Comprehensive empirical evaluation on both synthetic and real data
- Handles practical challenges like partial observability and sparse measurements

Weaknesses:

- My main concerns were about scalability, which is not necessary for a proof of concept publication as intended here.

---

> ### Author Response · Authors · 2025-02-07
>
> We thank the reviewer for the very positive feedback and the constructive comments, which we address below.
>
> Even though this is left as a future work, I would still like to see at least a simple example showcasing how the computational complexity would scale for higher dimensional systems? Even a simple back of the envelope calculation would be valuable.
> The runtime comparison suggests SDFL is significantly slower than baselines. Are there opportunities for optimization?
>
> $\longrightarrow$ We have added a remark in subsection 5.3 discussing that. We give an estimate of the increase in the computational time, as multiplied by a factor $q^d$ where $q$ is the size of the set of elementary functions and $d$ the added dimensions. Concretely, if we consider a system of dimension 5 instead of 3, the computational time of SDFL would approximately be multiplied by 50.
>
> $\longrightarrow$ As for opportunities for optimization, there are indeed at least two of them: one is to parallelize the stochastic roll-outs and the other is to parallelize the runs of different episodes. This can be done easily given the structure of the scheme.

---

### Review · Reviewer_JcUL · 2025-01-31

**Summary Of Contributions:**

This paper introduces a method called Symbolic Distribution Flow Learner (SDFL) to learn equations underlying probabilistic network flows. Unlike the rather recent, black-box ODE inference techniques, SDFL is a white-box approach that uses symbolic search to recover differential equations. SDFL employs a Wasserstein-based loss function that is compatible with probabilistic real-world systems. The authors also provide a sample complexity result, quantifying the number of required snapshots for reliable model discovery. The findings on Kuramoto networks and single-cell RNA sequencing trajectories show the efficacy of the presented approach.

**Audience:**

Yes

**Broader Impact Concerns:**

-

**Claims And Evidence:**

Yes

**Requested Changes:**

### Additional experiments
- Comparison against standard neural ODE baseline, which is actually a probabilistic approach as the initial value is sampled from a distribution.
- Model parameters/assumptions need to be ablated (initial value distribution, problem dimensionality, $n$, $m$, $L$, etc.)
- It would be interesting to see how the model performs if we had larger set of unitary and binary expressions, e.g., absolute value and powers.
- I would also like to see if the estimation would still be fine if we had a different Kuramoto network (with different $K$s.)
- An analysis on how the method scales to high-dimensional spaces is necessary. Now, it seems that the experiments are conducted only on 3D systems. What would be the computational cost of the Wasserstein loss in, e.g., 10 or 50 dimensions?

### Presentation
- It would be great to have a conceptual/summarizing figure early in the text. Figure 1 and 6 are the candidates but it is not clear to me what they communicate.
- Algorithm 1 can be detailed, I find it too generic now
- Does proposition 1 apply the presented method or is it a generic one?
- Beginning of section 4 can be written more verbosely, it is now difficult to follow (for example, terms that are not defined are used.)
- It would be good to see how exactly the loss is computed as it is rather abstract now.
- In Fig3, what does the "Euclidean distance" refer to? Is it computed after training or is it a way of measuring the difference between the estimated trajectories and ground truth?

**Strengths And Weaknesses:**

### Strenghts
- The motivation of the work, its scope, as well as the writing is clear. I also think it is a timely problem since the limits of black-box function estimators for ODE systems is already reached and white-/grey-box approaches have a lot of potential to explain system dynamics.
- Using Wasserstein distance (rather than the standard MSE or evidence lower bound based losses) for loss computation is a solid idea, given that the aim is modeling probability distribution flows.
- I did not see any flaw in the presented approach although some details are to be clarified.

### Weaknesses
- The biggest weakness is the limited experimental setup: ablations and comparisons with more methods are needed (see the next section). These additional experiments are not to show the superiority of SDFL but to see if benchmarking is done right (in other words, to see the robustness and limits of the method).
- Presentation needs to be improved (see the next section again)
- Unless I'm missing anything, the estimation of initial value is not described. Also, to my understanding, it is the only source of uncertainty. I would be happy to hear from authors how they estimated it and how it impacts the overall performance.

### Minor comments/weaknesses/questions
- I am not sure if the presented approach is suitable _in particular_ to network inference. It seems to me as a quite generic approach; hence, I do not understand why authors focused on "network modeling". If the focus is indeed on network modeling, then I would be glad to see experiments on more complicated networks
- What is the main message of theorem 5.1? As I understand, it justifies the use of a discrete loss function to estimate a continuous-time system but I guess this is the only possible way to estimate the system?
- What is the interpretation of the equations inferred in the single-cell population dynamics section?
- The reported losses/errors are in Wasserstein distance, which is the objective the model minimizes. What are the corresponding MSEs (or are they the values reported in Tables 5-9)?
- Concerning figure 5: if we only look at state trajectories, then neural ODEs are also equally informative. I feel the merits of this approach lie in the estimated closed-form equations, which authors can demonstrate better

---

> ### Author Response · Authors · 2025-02-07
>
> We thank the reviewer for the positive feedback and constructive comments, which we address below.
>
> I am not sure if the presented approach is suitable in particular to network inference. It seems to me as a quite generic approach; hence, I do not understand why authors focused on "network modeling". If the focus is indeed on network modeling, then I would be glad to see experiments on more complicated networks.
>
> $\longrightarrow$ We agree with the reviewer that the scope of the work is not limited to networks, and to better reflect his remark we have changed the title of the paper.
>
> What is the main message of theorem 5.1? As I understand, it justifies the use of a discrete loss function to estimate a continuous-time system but I guess this is the only possible way to estimate the system? \\
>
> $\longrightarrow$ Indeed, that would be the default implementation, however, it is insightful to know how quickly the discrete loss approaches the idealized one, since the latter has implications on uniqueness notably. Also please note that other discrete loss functions could be utilized, e.g., by leveraging kernels, yet to keep the study focused we only considered the standard discretization.
>
>  What is the interpretation of the equations inferred in the single-cell population dynamics section?
>
> $\longrightarrow$ Please note that this is included in the explainability Section 6.  In short, the inferred equations hint that the process has an oscillatory component, which a purely linear approximation would fail to capture.
>
>
>  The reported losses/errors are in Wasserstein distance, which is the objective the model minimizes. What are the corresponding MSEs (or are they the values reported in Tables 5-9)?
>
> $\longrightarrow$ We thank the reviewer for the question. Please note that the considered setting  (which is motivated by practical problems such as single-cell RNA sequence modeling) generally does not permit the direct computation of the Euclidean distance. This is because we observe discrete probability distributions which are not paired, i.e. they do not have the same support. As an example if we consider the probability measures/distributions  $\mu_1 =  \sum_{i=1}^n p_i \delta_{x_i}$ and $\mu_2 =  \sum_{j=1}^n q_j \delta_{y_j}$, we observe that the Euclidean distance between $\mu_1$ and $\mu_2$ can not be computed. In fact, even the KL divergence cannot be computed unless $( x_i, \, i \leq n ) = (y_j, \, j \leq n  )$.
> Hence, in this setting, and following previous works [1, 2, 3, 4], we conduct the evaluation based on the Wasserstein distance.
>
>
>
> [1] C. Bunne, L. Papaxanthos, A. Krause, and M. Cuturi. Proximal optimal transport
> modeling of population dynamics. In International Conference on Artificial Intelligence
> and Statistics, pages 6511–6528. PMLR, 2022.
>
> [2] L. Chizat, S. Zhang, M. Heitz, and G. Schiebinger. Trajectory inference via mean-field
> langevin in path space. Advances in Neural Information Processing Systems, 35:16731–
> 16742, 2022.
>
> [3] G. Huguet, D. S. Magruder, A. Tong, O. Fasina, M. Kuchroo, G. Wolf, and S. Krish-
> naswamy. Manifold interpolating optimal-transport flows for trajectory inference. Ad-
> vances in Neural Information Processing Systems, 35:29705–29718, 2022
>
> [4] A. Tong, J. Huang, G. Wolf, D. Van Dijk, and S. Krishnaswamy. Trajectorynet: A
> dynamic optimal transport network for modeling cellular dynamics. In International
> conference on machine learning, pages 9526–9536. PMLR, 2020

---

> ### Author Response · Authors · 2025-02-07
>
> Concerning figure 5: if we only look at state trajectories, then neural ODEs are also equally informative. I feel the merits of this approach lie in the estimated closed-form equations, which authors can demonstrate better. \\
>
> $\longrightarrow$ We agree with the reviewer that one of the main merits of the approach lies in the closed form nature of the model inference. However please note that in Fig. 5, we observe an error reduction achieved by SDFL. The inferred trajectories by SDFL turn out to be closer to the ground truth.
>
>
>
>
> Requested Changes:
>
> Comparison against standard neural ODE baseline, which is actually a probabilistic approach as the initial value is sampled from a distribution.
>
> $\longrightarrow$ We thank the reviewer for the comment, however please note that the state-of-the-art competitor TrajectoryNet is a neural ODE method, and this comparison is already reported.
>
>
>  Model parameters/assumptions need to be ablated (initial value distribution, problem dimensionality, $n, m, L, $ etc.)
>
> $\longrightarrow$ Please note that the considered setting (which was also considered in previous works [1, 2] assumes the initial condition to be known as well as a few screenshots across time. The point is to infer a dynamic model that captures the essence of the underling process, which is only sparsely measured across time.
>
>  $\longrightarrow$ As for $n$ and $m$, they represent respectively, the number of screenshots and the number of datapoints per screenshot. The experimental results we report already consider varying sample size, meaning $m$. As for the number of screenshots $n$, we have explored in Subsection E.4., in the context of comparison against symbolic regression state-of-the-art. Additionally, we also explore ablating the loss, since using the euclidean distance is equivalent to running MCTS as proposed by [3].
>
> It would be interesting to see how the model performs if we had larger set of unitary and binary expressions, e.g., absolute value and powers.
>
> $\longrightarrow$ Please note that powers are included by default since, multiplication of variables is included. As for the absolute value, it is a non-smooth function which is excluded by design. Extending the framework to non-smooth functions is an interesting aspect for future work though.
>
> I would also like to see if the estimation would still be fine if we had a different Kuramoto network (with different Ks.)
>
> $\longrightarrow$ We thank the reviewer for the comment. We have added such an experiment in subsetion E.4 of the Appendix.
>
> An analysis on how the method scales to high-dimensional spaces is necessary. Now, it seems that the experiments are conducted only on 3D systems. What would be the computational cost of the Wasserstein loss in, e.g., 10 or 50 dimensions? \\
>
> $\longrightarrow$ We have added a discussion of Wasserstein loss computation in Subsection 4.2. The scaling is dominated by the number of samples which is in $O(n^3)$, while it is linear in the dimensions of the system. We have also added a discussion of how the computational time of the whole scheme would scale for higher dimensions in Subsection 5.3
>
> [1] C. Bunne, L. Papaxanthos, A. Krause, and M. Cuturi. Proximal optimal transport
> modeling of population dynamics. In International Conference on Artificial Intelligence
> and Statistics, pages 6511–6528. PMLR, 2022.
>
> [2] A. Tong, J. Huang, G. Wolf, D. Van Dijk, and S. Krishnaswamy. Trajectorynet: A
> dynamic optimal transport network for modeling cellular dynamics. In International
> conference on machine learning, pages 9526–9536. PMLR, 2020.
>
> [3] . Sun, Y. Liu, J.-X. Wang, and H. Sun. Symbolic physics learner: Discovering governing
> equations via monte carlo tree search. In The Eleventh International Conference on
> Learning Representations, 2023

---

> ### Author Response · Authors · 2025-02-07
>
> It would be great to have a conceptual/summarizing figure early in the text. Figure 1 and 6 are the candidates but it is not clear to me what they communicate. Algorithm 1 can be detailed, I find it too generic now
>
> $\longrightarrow$ Please note that we reported a detailed version of the algorithm in Appendix D
>
> Does proposition 1 apply the presented method or is it a generic one?
>
> $\longrightarrow$ Proposition 1 is indeed specific to the presented method.
>
>  Beginning of section 4 can be written more verbosely, it is now difficult to follow (for example, terms that are not defined are used.)
>
> $\longrightarrow$ Thanks for making this point, we have added more explanations.
>
> It would be good to see how exactly the loss is computed as it is rather abstract now.
>
> $\longrightarrow$ We have added how the loss function is computed concretely in Subsection 4.2
>
> In Fig3, what does the "Euclidean distance" refer to? Is it computed after training or is it a way of measuring the difference between the estimated trajectories and ground truth?
>
> $\longrightarrow$ In Fig. 3, Euclidean distance is indeed computed after training. We have clarified this in Subsection 5.1

---

### Comment · Action_Editor_SxNF · 2025-03-26
**Camera-ready**

Dear authors,

It appears your submission is still using the "Under review" template.
Please update your submission according to the camera-ready instructions.

Best,
AE

---

### Decision · Action_Editor_SxNF · 2025-03-07

**Recommendation:** Accept as is

**Comment:**

Reviewers were unanimous in recommending acceptance of this work and I agree with this assessment.

**Audience:**

Yes, there is an audience.

**Claims And Evidence:**

No concern with the claims and evidence were raised by reviewers.